# A Hybrid Two-Axis Force Sensor for the Mesoscopic Structural Superlubricity Studies

**DOI:** 10.3390/s19153431

**Published:** 2019-08-05

**Authors:** Taotao Sun, Zhanghui Wu, Zhihong Li, Quanshui Zheng, Li Lin

**Affiliations:** 1Center for Nano and Micro Mechanics, Tsinghua University, Beijing 100084, China; 2State Key Laboratory of Tribology, Tsinghua University, Beijing 100084, China; 3Department of Engineering Mechanics, Tsinghua University, Beijing 100084, China; 4National Key Laboratory of Science and Technology on Micro/Nano Fabrication, Institute of Microelectronics, Peking University, Beijing 100871, China

**Keywords:** multi-axis force sensor, hybrid design, high resolution and long range, real-time observation, mesoscopic structural superlubricity

## Abstract

Structural superlubricity (SSL) is a state of nearly zero friction and zero wear between two directly contacted solid surfaces. Recently, SSL was achieved in mesoscale and thus opened the SSL technology which promises great applications in Micro-electromechanical Systems (MEMS), sensors, storage technologies, etc. However, load issues in current mesoscale SSL studies are still not clear. The great challenge is to simultaneously measure both the ultralow shear forces and the much larger normal forces, although the widely used frictional force microscopes (FFM) and micro tribometers can satisfy the shear forces and normal forces requirements, respectively. Here we propose a hybrid two-axis force sensor that can well fill the blank between the capabilities of FFM and micro tribometers for the mesoscopic SSL studies. The proposed sensor can afford 1mN normal load with 10 nN lateral resolution. Moreover, the probe of the sensor is designed at the edge of the structure for the convenience of real-time optical observation. Calibrations and preliminary experiments are conducted to validate the performance of the design.

## 1. Introduction

Although the concept of structural superlubricity (SSL) dates to 1983 [1], from the first SSL observation in 2004 [2] to date, experimental studies of SSL have still been mostly in the nanoscale [3,4,5,6,7]. However, for practical applications in most cases, the contact scale needs to be much larger. The first microscale SSL observation is reported in 2012 [8]. Follow that, studies on large scale SSL (mesoscale: One dimension of the system is from 1 μm to 1 mm) have attracted much attention [9,10,11,12,13]. Even with that success, friction behavior with respect to different loading conditions in current mesoscale SSL studies is not clear, which causes other crucial problems towards practical applications [6,11,13,14].

The major challenge in the mesoscale SSL load studies is to simultaneously acquire both the ultralow shear force and the much larger normal force, since both of the forces typically occur in mesoscale SSL at a time. Currently, the two widely used tools for tribology studies are frictional force microscopes (FFM, a kind of AFM (Atomic Force Microscope) specialized of the tribology) and tribometers. However, both cannot simultaneously satisfy the requirements. The resolution of FFM is ultrahigh, which is easy to achieve sub-nanonewtons. However, the maximum normal load could be applied by FFM is highly depends on the properties of probes. Most of the commercial AFM probes are designed for the purpose of small force detections, which have a normal stiffness in the order of 0.1–100 N/m and correspond to the normal load approximately from <1 nN to 100 μN [13,15]. On the other hand, force apparatus in the tribometer, which is designed for the macro friction studies [16], can easily achieve loads larger than millinewtons, but only with a usable resolution in tens of micronewtons [17], due to the macro structures. This cannot meet the requirements to detect small shear forces in sub-micronewtons in mesoscale SSL (Judged by the coefficient in SSL state, which is below 0.001 [18]). Figure 1 gives the comparison of the force requirements for SSL studies and the force ranges of the existing instruments. It indicates that current instruments cannot fully cover the whole force range for the SSL studies, especially for the high load mesoscale SSL studied.

Besides the commercial instruments developed specifically for tribology studies, some individual force sensors may offer additional possibilities. In the early studies of the mesoscale SSL, a commercial MEMS single-axis micro force sensor was used [19]. However, the single-axis configuration can only measure tangential forces without normal forces applied, therefore, it is not suitable for further friction studies. On the other hand, among many multi-DOF (Degree of Freedom) MEMS force sensors reported [15,20,21,22], there are still some drawbacks for the applications in the mesoscale SSL studies, such as the frangibility of the MEMS structures and the visualization problem due to the position of the tip [2]. Recently, many works focusing on the modification of AFM probes have been reported. M. Michalowski et al. [23] introduced an idea to fabricate millimeter size AFM cantilevers using stainless steels; Lydéric Bocquet et al. [24] set up a two-axis measuring system using the tuning fork principle in qPlus AFM [25]; and N. T. Garabedian et al. [26] described a general method to bridge the gap between the nanotribology and the macrotribology by using the colloidal probe technique in AFM. However, problems are dynamic mode sensors have complicated physical relations between signals and forces, and it is impossible to observe the contact mates when using the colloidal probe technique because the cantilever always blocks the light. So far, there is still a great desire to have an appropriate tool for the high load mesoscale SSL explorations.

In this paper, a novel design has been proposed for a single apparatus to simultaneously measure large normal forces and tiny shear forces to fill the instrumental gap for the high load mesoscale SSL studies. In addition, the probe of the sensor is at the edge of the structure for the convenience of real-time optical observation, and the issue of the structural robustness has also been taken into consideration. Although the work discussed here is for the applications in the SSL studies, the proposed idea that expanding the force range across multi-axis can be applied in other fields. The structure of the paper is: Section 2 describes the design principle of the sensor, followed by the analysis of the hybrid design in Section 3. In Section 4, calibration results and friction experiments are to demonstrate the performances of the sensor. Discussions for the current design are given in Section 5. Conclusions are drawn in Section 6.

## 2. Design Principles

### 2.1. The Hybrid Concept

Usually, materials tend to break under the stress of GPa. To reach a high contact pressure over 1 GPa (average stress) on the mesoscale contact area (1 μm × 1 μm for example), the normal load needs to reach at least 1 mN. Meanwhile, the friction of the SSL sample under such high load could be smaller than 1 μN if the superlubricity state is maintained. That means the requirement of the resolution in the tangential direction is 100 nN or even smaller, resulting in at least four orders of magnitude of force span between the normal and tangential directions.

Force sensors with the high resolution in the sub-micro newton level can be accomplished by taking advantages of the micro fabrication techniques (MFT), such as lithography. Different from the traditional macro machining methods using cutting tools, MFT is good at fabricating structures in microscale with very high accuracy. This greatly decreases the stiffness of the sensing structures to achieve high sensitivities. In addition, small structures are always insensitive to the environment disturbance; thus, can further improve the system signal-noise ratio. However, micro sensors fabricated by MFT are usually not suitable to afford large forces and have disadvantages of difficult in 3D design, complicated and costly in early development and really fragile in use. Moreover, force sensitivities are usually comparable in each direction in commonly used multi-axis force sensors. This situation limits the force span across different axes. Therefore, new strategies are needed to make a design with a large span of forces and a robust structure.

Here, we propose a hybrid design. The idea is combining the macro-fabrication structure with the micro-fabrication structure in the high load and high resolution directions, respectively, to form a two-axis sensor. The hybrid design has a large difference of stiffness between different sensing directions, which can greatly increase the force span across axes. The design with rigid macro structures in the high load direction is more robust compared to the traditional MEMS sensors, since traditional MEMS sensors are purely consisted of micro-fabricated fragile structures. However, in a multi-axis sensor, all structures have to withstand forces from all the directions inevitably. It means that the micro structure we designed for the high resolution has to bear large loads in the normal direction. So a robust micro structure has been designed to mitigate the frangibility problem, while pursuing the high sensitivity at the same time. The schematic model of the hybrid two-axis force sensor is shown in Figure 2. The sensor is designed to detect large normal forces in the Z axis and small lateral forces in the X axis. Since the micro structure is very small, a macro adapter board is designed to carry the micro structure, bring signals out and assist assembly of the two-axis sensor.

### 2.2. Sensor Design

Among various force sensing structures, the double-(leaf-)cantilever is widely used for force sensing, because it only produces bending deformations at the free end when normal forces are applied [27]. Without rotations of the free end, the sensing planes are kept parallel during the translation movements, which greatly benefits the accuracy of measurements. We apply the double-cantilever structure for our two-axis sensor in the normal load direction. To further improve the stability of the double-cantilever structure, both upper and lower leaves are split into two parts and separated for a distance. This increases the rigidity against the rotation about the long-axis and makes the structure more immune to assembly errors (details about assembly issues are provided in Section 3). A 3D model of our designed double-cantilever pair is provided in Figure 3a. As the deformation of the structure during measurements is small, the stiffness of the double-cantilever pair can be simply calculated from the equation:(1)$$K_{N}=\frac{N}{d_{z}}=\frac{2Ewt^{3}}{l^{3}},$$
where $K_{N}$ is the normal stiffness of the whole structure, N is the normal force, $d_{z}$ is the normal deflection at the free end, E is the elastic modulus of the material, w, t and l are the width, thickness and length of each beam respectively. The material we used for the cantilever leaves is the stainless steel with E=200 GPa, fabricated through the precision wire electrical discharge machining. One example of the design parameters is listed in Table 1. The calculated stiffness, in this case, is 127.55 N/m.

Among various measurement methods, the capacitance principle is widely used in high performance situations [28] because it has high sensitivity and is less dependent on temperatures. In our design, a differential, variable, capacitive configuration is used, which provides electronic signals from applied forces. The differential configuration makes the signal almost immune to common environmental changes by subtracting the effects of noises on the two similar capacitors, which is beneficial for operations in the general atmosphere. As shown in Figure 3b, the double-cantilever pair acts as a mover, and two variable capacitors are formed between the surfaces of the beams and the electrodes on the upper and lower supporters, respectively. Deformations of the double-cantilever pair change the gaps of the two capacitors oppositely. Since there are only translation movements in the normal direction for the designed cantilever, the relationship between the structural deformation $d_{z}$ and the differential capacitive signal $C_{N}$ can be directly written as:(2)$$C_{N}=C_{N1}-C_{N2}=\frac{2\varepsilon Ad_{z}}{d_{z0}^{2}-d_{z}^{2}},$$
where $C_{N1}$ and $C_{N2}$ are the capacitive signals in the two opposite moving directions, ε is the dielectric constant of the air, A is the area of the capacitor, $d_{z0}$ is the initial gap of the capacitor structure. After rearranging Equation (2) with Equation (1) and utilizing the Taylor series expansion near $d_{z}=0$, the relationship between the measurement force and the capacitive signal can be approximately expressed in a linear format:(3)$$N=\frac{d_{z0}^{2}}{2\varepsilon A}\cdot K_{N}\cdot C_{N}.$$

In the lateral direction, due to the demanding requirement of the sub-micronewton resolution, the differential capacitor configuration is also employed to pursue a better sensitivity. In addition to this, the MFT is adopted to achieve the high resolution. Different from the most-used micro comb array structures in capacitive force sensors [20,21,22], we applied a teeter-totter structure [29,30,31] for the lateral force detections. Although the capacitive force sensors utilizing differential comb structures have been reported to reach high resolution in nanonewtons, it is lack of sturdiness. Although stoppers are set along the sensitive directions to prevent the overload breaks, the springs and combs of the sensor can easily be destroyed by forces out of the sensing plane. Besides, large areas of the comb array exposing to the atmosphere is likely to be struck by the environmental contaminants and finally causes structural failure. The teeter-totter structure, on the other hand, is much more robust. The structure of the teeter-totter structure can be seen from Figure 3c,d. It consists of one mover with two supporting beams (Figure 3c). The ends of the beams extend out like feet to stand firmly on the base board, remaining a small gap between the mover and the board (Figure 3d). There are two electrodes on the base board, and one tip on the lower part of the mover. Forces acting on the tip cause the mover to rotate about the two supporting beams, thus producing angle changes. During measurements, the base board acts as a stopper, which can well prevent the break caused by the overload. In addition, the structure is concise to have fewer possibilities for pollution caused failure problems. When applying lateral forces on the tip, there are three primary deformation formats in the teeter-totter structure: The torsional deformation about the supporting beams, the collapsing deformation along the beams and the bending deformation of the lower part of the mover plate (see Appendix A). However, our analysis indicates that the torsional stiffness is two orders of magnitude smaller than the other two stiffness. Therefore, the system stiffness of the teeter-totter structure can be expressed simply as the torsional stiffness of the supporting beams(details are provided in Appendix A and also in ref. [31]). The torsional stiffness of the teeter-totter structure is:(4)$$K_{t}=\frac{T}{\theta }=\frac{2G\beta wt^{3}}{l},$$
where T is the moment act on the structure, θ is the angular displacement of the mover plate, G is the shear modulus of the material, β, w, t, l are the geometric parameters of the supporting beam.

In order to compare the stiffness between the two sensing directions, we transfer the torsional stiffness expression of the teeter-totter structure into the linear format in Equation (5). For tiny rotational angle (−0.005 rad), the formula of the moment can be simplified as T=FL, where F is the tangential force applied on the sensing point, L is the distance between the sensing point and the torsional axis (Figure 3d). The displacement at the sensing point can be written as $d_{x}=\theta L$. The linear stiffness expression of the double cantilever becomes:(5)$$K_{F}=\frac{F}{d_{x}}=\frac{2G\beta wt^{3}}{lL^{2}},$$

Here, we also list the other two equations showing the expression of the rotation angle θ and the relationship between the torsional stiffness $K_{t}$ and the linear stiffness $K_{F}$ by rearranging Equations (4) and (5). Both will be used in later studies.
(6)$$\theta =\frac{FL}{K_{t}},$$
(7)$$K_{F}=\frac{K_{t}}{L^{2}}.$$

For a wider range of applications, we have designed different beam sizes. All the parameters are shown in Table 1. The material used in the micro fabrication process is silicon with a shear modulus of 62 GPa. The calculated linear stiffness of the teeter-totter structures is 1.30–14.9 N/m. Compared to the stiffness in the normal direction, our hybrid configuration has a stiffness difference in the order of 10–100 between the two sensing directions. The micro fabrication process of the teeter-totter structure has been described in details in our early work [31].

The teeter-totter structure is widely used in projectors as micro-mirrors [32] and in robots as tactile sensors [30], for that its symmetric structure is beneficial for the applications in the differential capacitance measurement. As can be seen from Figure 3d, two capacitors are formed between the mover plate and the two electrodes on the fixed base, respectively. Rotation of the mover changes the gaps of the two capacitors oppositely. With the Taylor series expansion near θ=0, the relationship between the angular displacement and the differential capacitive signal can be expressed as:(8)$$C_{L}=C_{L1}-C_{L2}=\frac{\varepsilon Al}{d_{x0}^{2}}\cdot \theta ,$$
where $C_{L1}$ and $C_{L2}$ are the capacitive signals in the two opposite moving directions, $d_{x0}$ is the initial gap of the teeter-totter capacitor. After rearranging Equation (8) with Equations (6) and (7), the relationship between the measurement force and the capacitive signal can be written as:(9)$$F=\frac{d_{x0}^{2}}{\varepsilon A}\cdot K_{F}\cdot C_{L}.$$

In summary, measurements of the differential capacitive signals $C_{N}$ in Equation (3) and $C_{L}$ in Equation (9) provide complete information of forces in the normal and tangential directions required for the friction studies.

## 3. Analysis of the Hybrid Design

### 3.1. Strength of the Micro Structure

During the usage of a multi-axis force sensor, each part has to withstand forces in the undesired measurement directions. The major consideration is that the micro teeter-totter structure has to withstand much larger forces in the normal direction while measuring small tangential forces. Besides the torsional deformation caused by the tangential forces, a bending deformation also acts on the teeter-totter beams because of the normal loads. Here, we use the finite element analysis (FEA) to simulate the stress state of the micro teeter-totter structure under the normal load of 1 mN and the lateral force of 100 μN. It is regarded as the max load based on the design specifications and the possible displacement range of the teeter-totter mover. The max equivalent stress calculated by the FEA for the #1 beam which has the smallest stiffness is 48.72 MPa (Figure 4), much smaller than the yield stress 2–7 GPa of the silicon material. Therefore, our designed micro structure is reasonably competent in the detections under large loads. (Strength of the extending probe is analyzed in Appendix A. The calculation results are displayed in Appendix A.)

### 3.2. Coupling of the Signals

Coupling is defined as the crosstalk of a sensor between different sensing directions. It exists in all multi-axis force sensors, but is usually undesirable because it results in measurement errors and complicates the signal process. The coupling may come from an inappropriate stiffness in the undesired sensing direction, geometric correlations between moveable structures, or may arise from other physical interacts. Analysis indicates that there is no obvious coupling in our hybrid sensor, due to the stiffness problems and the capacitive signal interactions between the two axes. The major coupling problem is caused by geometric issues. (Details of coupling analysis are given in Appendix A. The corresponding results are shown in Appendix A).

The geometric coupling property in our configuration is just like that in the AFM cantilevers, which also combines a torsional and bending deformations in one system [33,34,35]. Geometric illustrations are given in Figure 5. A side view of the teeter-totter structure with a tip glued on the bottom part of the mover plate is shown in Figure 5a. An exaggerated angle is drawn to show the angle (3°) formed between the tip and the mover plate after the assembly of the tip. In order to express the geometric relationship clearly, a simplified model is made in Figure 5b where the fixed base is omitted, and the angle is represented by an equivalent angle α. When forces act on the tip, the lateral signal measured by the teeter-totter sensor is a representative of the angle between the mover plate and the fixed base, which is denoted as θ in Figure 5c. As can be seen from the two common situations illustrated in Figure 5c, the change of θ is a superposition of the friction force f to be measured, the normal load N and the gravity of the tip $G_{tip}$. The relation between the angle θ and the three external forces is:(10)$$fLcos\left(\theta +\alpha \right)+NLsin\left(\theta +\alpha \right)-\frac{1}{2}G_{tip}Lsin\left(\theta +\alpha \right)=K_{t}\theta ,$$
where *L* is the distance between the tip apex and the torsional axis, $K_{t}$ is the torsional stiffness of the teeter-totter beam described in Equation (4).

After rearranging Equation (10) using Equation (6) and small-angle approximations, the relation between the measured lateral forces F and the practical friction forces f can be expressed as:(11)$$F=\frac{K_{t}}{K_{t}+\frac{1}{2}GL-NL}\cdot f-\frac{K_{t}\left(\frac{1}{2}\alpha G-\alpha N\right)}{K_{t}+\frac{1}{2}GL-NL}.$$

As can be seen from Equation (11), the derivative of F with respect to f is a function of the normal load N, which represents an obvious coupling effect in the teeter-totter structure. Here, we define △F/△N under a constant friction force as a coupling rate to make a quantitative description of the coupling effect. The reason for the condition of f=const. is given in the Appendix A. Through calculation, the coupling rates are 4.98%, 8.36% and 4.79% for the #2, #3 and #5 teeter-totter structures respectively (experimental results can be seen in Section 4.2).

Friction forces f may be extracted from the Equation (11) by carefully calibrating all the parameters. However, the process is complicated and contains many sources of errors. So we employ another method to extract friction forces. The method is widely used in the FFM, where friction forces can be extracted from the friction loops enclosed by the trace and retrace curves (see experimental curves shown in Section 4.3). By subtracting Equation (10) in the two moving directions depicted in Figure 5c, we get an incremental form of equation:(12)$$\left(f_{+}+f_{-}\right)L+NL\left(△\theta_{+}-△\theta_{-}\right)-\frac{1}{2}G_{tip}L\left(△\theta_{+}-△\theta_{-}\right)=K_{t}\left(△\theta_{+}-△\theta_{-}\right),$$
where ‘+’ and ‘−’ represent forward and backward directions in Figure 5c, respectively, △θ is the incremental form of the angle θ. Compared to Equation (11), the angle α is eliminated in Equation (12). The incremental form of angle avoids the definition of the initial angle $\theta_{0}$ ($\theta =\theta_{0}+△\theta$), which is not easy to obtain.

We assume that the friction forces within one friction loop are the same. The relationship between the measured lateral forces and the practical frictional forces is given as:(13)$$f=A\left(N\right)\cdot \frac{1}{2}\left(F_{+}-F_{-}\right),$$
where, A(N) is a correction factor we denoted to replace the expression $1+\left(G_{tip}/2-N\right)L/K_{t}$. Through calculation, the correction factor A(N) is between 0.956–1 within the normal load of 1mN. It indicates that the error will be no more than 5% without making the corrections.

### 3.3. Assembly Errors

Assembly is an important step in the construction of the hybrid structure. Since our hybrid sensor needs to assemble separate parts manually, we must consider the influence of the assembly errors. The errors are analyzed from three projection planes as depicted in Appendix A. The analyses indicate that through structural optimization, the signals are immune to most of the assembly errors (error < 1%). In our hybrid sensor, the effect of the assembly error is most likely to be reflected by the stiffness variation of the double-cantilever structure. To further demonstrate this effect, calibration experiments are conducted in the later Part (Section 4.2). More details can be seen in the Appendix A).

## 4. Performance

### 4.1. The Test System

A photo of the hybrid two-axis force sensor is given in Figure 6a. An scanning electron microscope (SEM) picture shows the micro teeter-totter structures (see Figure 6b). Capacitive signals in the two sensing directions are read out separately by a control system developed by ourselves. All the components in the system are commercially available. The system has two measurement channels reading two pairs of differential capacitive signals simultaneously. Each capacitive signal is directly converted to a digital signal with a 24-bit resolution at a maximum sampling rate of 10 kHz. Besides the capacitive signal acquisition part, the control system also has one direct current (DC) voltage output channel and a Microcontroller Unit (MCU). In the MCU, a proportional–integral–derivative (PID) controller is built. Real-time normal force signals are sent into the PID controller, compared to a target normal load to generate error signals. The error signals are converted to the DC output voltages to adjust the displacement of the connected actuator. The voltage range of the DC output is 0–100 V, with an 18-bit resolution.

In order to demonstrate performances of the sensor, a test system is built. A photo of the established test system can be seen in Figure 6c. The hybrid sensor is attached to a vertical moving stage of the system. The vertical stage has coarse and fine two-level adjustments. The fine stage employs a home-built piezoelectric actuator. It is closed-loop controlled to keep a constant target load in the normal direction with a resolution in the order of 10 μN. The lateral moving stage, located under the hybrid force sensor, is used to place samples and make lateral scans. The lateral stage also has both coarse and fine adjustments. The fine stage is a commercial nano-positioning stage (MCL) with a resolution of 0.4 nm. Apart from the force sensor and the moving stages, a commercial optical microscope (Hirox, Tokyo, Japan) is set in the system for the real-time observation. Besides, a customized LabVIEW interface is developed on a PC for user interactions.

### 4.2. Sensor Calibrations

A commercial high precision balance (METTLER TOLEDO, 10 μg resolution, 81 g range, Greifensee, Switzerland) is applied to calibrate the double cantilever structure. In the calibration experiments, the double cantilever is fixed on the vertical moving stages, as shown in Figure 6c. During calibrations, the piezoelectric actuator stretches out to make the free end of the double cantilever contact with the balance. Both digital differential capacitive signals and forces from the balance are recorded. Calibration results are given in Figure 7a. The signal shows good linearity within the usable force range of 1.5 mN. The inserted picture in Figure 7a displays a period of peak-to-peak noise signals. We define the resolution of the sensor by the peak-to peak magnitude of the noise signal, which is 40 μN is this case. Besides, the stiffness of the double cantilever is also calibrated. The result is 150.59 N/m, slightly larger than the result calculated from Equation (1) which is 127 N/m. This may be because of the errors from the macro fabrication and the calibration experiment.

The teeter-totter sensing part is calibrated by using a commercial micro-force sensor (FemtoTools, FT-S100, 5 nN resolution, ±100 μN range, Zürich, Switzerland). To study the influence of the extending probe, two kinds of calibrations are made. The first calibration is made at the edge of the mover plate. The experimental results show a torsional stiffness of 124.93 μN·m. It is larger than the calculated result 111.90 μN·m, shown in Table 1. The difference may mainly attribute to the fillets at the beam corners (see Appendix A in Appendix A) and may also come from the errors of the calibration position, fabricated geometric size or material parameters. Then, we calibrated the sensor at the tip. Before calibrations, a tungsten wire (diameter 0.3 mm) is electrochemically etched to form a sharp tip (radius 200 nm, observed by SEM) and bonded to the teeter-totter plate by the ultraviolet (UV)-glue. This time the experimental force stiffness is 3.20 N/m, slightly smaller than the calculated result 3.57 N/m, shown in Table 1. The decrease of the stiffness may possibly come from the deformation of the extending tip and the bounding glue. The results shown here are of the #2 teeter-totter structure; however, all the other teeter-totter structures display the same trend. Figure 7b shows the relationship between the capacitive signal and the calibrated force. From the figure we can see that the force range of the #2 lateral sensor is 30 μN. Beyond 30 μN the edge of the mover plate touches the supporting board, so the force increases rapidly without the change of the capacitive signal. On the other hand, this also demonstrates the robustness of the micro teeter-totter structure. As can be seen from Figure 7b, there is a considerable nonlinearity (nonlinear error = 9.1%) within the force range. It is because a Taylor series expansion is applied to get a linear relationship in Equation (8) and the linear expansion can only work well near small deflections. To increase linearity, a better range to use is 25 μN (nonlinear error = 6.5%). Judged by the peak-to-peak noise, shown in the inserted picture in Figure 7b, the resolution of the #2 teeter-totter structure is 10 nN. More calibration results of other teeter-totter sensing parts are summarized in Table 2.

Calibrations of the geometric coupling effect on the teeter-totter structure described in Section 3.2 is conducted by applying an ultralow friction system. The reason for choosing a system with very small friction is to reduce the variation of the friction force, thus to reduce the influence on the calibration of the coupling rate as is described in Section 3.2 and Appendix A. It has been reported that the magnetic levitation system has a very small friction coefficient [35,36], which is a promising choice. However, the magnetic field may influence the capacitance measurements, and the magnetic levitation system requires additional operations to calibrate. Therefore, we choose the SSL system which is reported having a nearly zero friction property to conduct the work. The SSL pairs we used is graphene-graphene. Although the ultralow friction between two graphene interfaces is only experimentally confirmed under small loads, our experiments show that the frictions under higher loads are still small enough to be ignored (the experimental curves can be seen in Appendix A). Figure 7c shows the comparisons between theoretical and experimental results of the coupling effect. The trend has good consistency, which approves our model described in Section 3.2.

As mentioned in Section 3.3, the assembly may bring some uncertainties of measurements. The possible repeatability problem of the signal caused by different assemblies can be primarily reflected in the stiffness variations of the double-cantilever. To further illustrate this issue, five groups of calibrations representing five times of assemblies are performed. Within each assembly, five repeat measurements are made. The results are shown in Figure 7d. Each height of the column represents an average calibrated force constant ($△N/△C_{N}$) of each assembly. By comparing the height of each column, one can study the variations caused by different assemblies. In addition, the error bar on each column is the standard deviation (std) of the force constants got from the five repeated experiments with the same assembly. This reflects the repeatability of the normal force measurement. As can be seen in Figure 7d, the std within the same assembly is in the range of 5.48 × 10^–6^–2.77 × 10^–5^ μN/signal, only has a 0.3% fluctuation of the average value. Meanwhile, the std between different assemblies is 4.96 × 10^–5^ μN/signal, displays a slightly larger fluctuation caused by the assembly (0.6%). However, the variation of the signal is still very small. Besides, there is one more thing to mention. The above experiments are done without connecting the wires between the two sensing structures, as is shown in Figure 6a. To further find the possible influence of the wires, calibrations with wires are conducted in another five sets of experiments (shown in gray-grid columns in Figure 7d). This time the std within the same assembly is in the range of 5.48 × 10^–6^–2.05 × 10^–5^ μN/signal with 0.3% fluctuation, the std between different assemblies is 3.27 × 10^–5^ μN/signal with 0.4% fluctuation.

### 4.3. Friction Experiments

Friction experiments are performed by using friction pairs from the SSL system. Samples are made by transferring the upper part of a self-retracted 5 μm × 5 μm graphite mesa onto a mica or a (multilayered) graphene flake. The self-retraction phenomenon and the transfer method of the graphite mesa can be seen from our previous works [8,37]. The system has a visualization advantage. As is shown in Figure 8a, the whole experimental procedure can be real-time monitored by an optical microscope.

Experimental results on a graphite-mica interface are illustrated in Figure 8b. It is measured by using the #3 teeter-totter beam. During the experiments, normal loads are applied in the sequence of 300 μN, 500 μN, 700 μN, 500 μN, 300 μN and 200 μN. Each load is repeated three times. As can be seen from the figure, the friction loops measured between the graphite—mica interface is in a clear shape. The loops are highly repeatable within the same load. One thing to note is that we did not shift the friction loops deliberately; the shifting comes from the geometric coupling effect. On the existence of the coupling, an obvious saturation of lateral signals occurs when the load reaches 700 μN, revealing a ceiling of the measurement. To further show some details of the frictional signal, we magnified a friction loop with the normal load of 200 μN in the bottom diagram of Figure 8b. The curve is displayed in lateral forces with respect to sample displacements. The area enclosed by the curve represents the energy dissipation of the friction loop. It can be seen from the result that lateral forces in different sliding directions are clearly distinguishable. Based on Equation (13), the friction force can be extracted by subtracting the two-directional lateral forces (forward and backwards). It is 11.12 μN in this case. In addition, friction forces under different loads are different, indicating a friction dependence of the load. The relationship between the normal force and the friction force is displayed in Figure 8d. The friction coefficient (△f/△N) calculated from the friction loops is 0.047.

The friction coefficient we measured on the graphite-mica interface is not small enough. It is consistent with previous works done by FFM, which also finds a higher friction force between the graphite–mica interface than that between the graphite-graphene interface. It is partly, due to the hydrophilic property of mica [38,39]. In order to verify the capability of the sensor for the ultralow friction studies, we perform another friction experiment on graphite-graphene pair (see Figure 8c). This time the friction loops are narrow, due to the ultra-small frictions between the two graphene surfaces. However, seen from the magnified loop at the bottom of Figure 8c, the lateral forces in the two sliding directions are still distinguishable. Moreover, the results also show a highly repeatability within each three repeated cycles. In this experiment, the #5 teeter-totter beam is applied for the lateral force measurement. Since the available normal range of the #5 teeter-totter structure summarized in Table 2 is larger than 1 mN, the maximum load applied in this experiment achieves 1 mN. The relationships between the normal force and the corresponding friction force is displayed in Figure 8d. Compared to the result of the graphite-mica interface, the friction coefficient of the graphite-graphene interface is much smaller, which is 0.0002 in this experiment. To further display the capability of our sensor to measure frictions, more experimental results are shown in Appendix A, Appendix A.

## 5. Discussion

Usually, multi-axis sensors are designed to have comparable stiffness in each measurement direction. A large difference of stiffness in different directions might cause some problem. However, for the study of SSL, especially on the mesoscale, forces in the normal and tangential direction required to have at least four orders of magnitude difference while covering the range from the macro down to the nanoscale. In order to overcome this challenge, we propose a hybrid design. The following strength, coupling and assembly issues are analyzed carefully to avoid possible failures caused by the large asymmetric structure feature.

The hybrid sensor has a coupling rate of around 5% between normal and lateral directions. However, although the coupling rate is small, the influence of the coupling is considerable, especially for the friction studies of SSL materials. Because the friction force always appears to be more than a thousand times lower than the normal load during the measurements. In the lateral direction, due to the existence of the coupling, the measurable range of the lateral force is reduced when applying a normal force, as mentioned in the graphite-mica experiment. It means under high loads, only samples with small friction coefficient can be measured. In the normal direction, on the other hand, the allowable range of the normal load is also limited by the available displacement range of the teeter-totter structure. Table 2 summarizes the maximum normal loads can be applied by each teeter-totter structure. As described in Section 3.2, the coupling rates caused by each teeter-totter structures have an obvious difference. The results show that the difference is not simply determined by the stiffness of the teeter-totter beam. As expressed in Equation (S7), the coupling rate is affected by the torsional stiffness ($K_{t}$) and parameters of the tip ($G_{tip},L,\alpha$). However, although the coupling effect is complicated, we studied a decoupling method by using friction loops. The results show that without special correction, the coupling error can be restricted within 5%.

For studies in the mesoscale, at least one dimension of the sample is always bigger than a micrometer which is visible with an optical microscope. Optical observation provides advantages of intuitive to understand phenomena and real-time to display what’s happening during experiments. In our design, the probe is set at the edge of the sensing structure so that it is convenient to locate a microscope without blocking the light for the real-time monitor. Benefitted by this in-situ observation, we can confirm the contact mates during movements and monitor the moving state during the whole friction process. It should be noted that the maximum magnification of the objective lens can be used in our system is 20×. This is restricted by the thickness of our hybrid sensor (the vertical distance between the tip and the upper surface of the support). The sensor structure can be improved to be more compact to further achieve a better magnification.

One of the drawbacks of the hybrid design is the need for manual assembly of the separating parts. It is time consuming and could bring some extra errors. Other than this, the macro structure has a relatively large weight which will lower the system dynamic. Through spectrum analysis by FFT, the natural frequency of the double cantilever structure is about 15Hz, which is consistent with the major frequency component of the noise signal. This indicates that the macro structure is sensitive to environmental mechanical vibrations. In order to suppress the effect of the vibration noises, a damper can be added in the sensor (see Figure 6a). Besides, a shorter beam can be designed, which will increase the system frequency. Further increase of the systematic dynamic property may introduce the electric feedback control [29,40] or other techniques, which might be done in the future.

## 6. Conclusions

In this paper, we develop a hybrid sensor. The sensor is designed to measure large normal forces and tiny shear forces simultaneously to fill the applicable force blank of the current friction test devices and to build a bridge for the SSL studies between the nano and macro scale. Experimental works are carried out to validate the performances of the sensor. The performances achieve the goals of our original design. Through calibration experiments, the range in the lateral force direction is ±30 μN–±170 μN with a resolution of 10 nN–80 nN, while the available normal load can be applied by the sensor can achieve more than 1 mN with a resolution of 40 μN. Between the two sensing axes, the sensor shows a three orders of magnitude difference in terms of resolution, and a five orders of magnitude force span which across from several tens of nanonewtons to several millinewtons. The maximum applicable normal load by our sensor is one order of magnitude higher than the highest load applicable by FFM. It rightly fills the instrumental gap between the FFM for the nanofriction studies and the tribometers for the macrofriction studies. This will assist us to bridge the gap between the nano and macro tribology. Additionally, the best friction result presents a capability of the sensor to measure the ultralow friction coefficient in the order of 0.0001, showing a great advantage for the mesoscale SSL studies. Besides, the sensor is designed for the convenience of real-time optical observation. Based on the current achievements, systematic studies on the mesoscale SSL will be done later. Apart from the advantages, drawbacks of the hybrid sensor, such as the geometric coupling and the error of assembly are also carefully studied. These studies will lead us to further instrumental improvements, which will be accomplished in our future works.

## Figures and Tables

**Figure 1 sensors-19-03431-f001:**
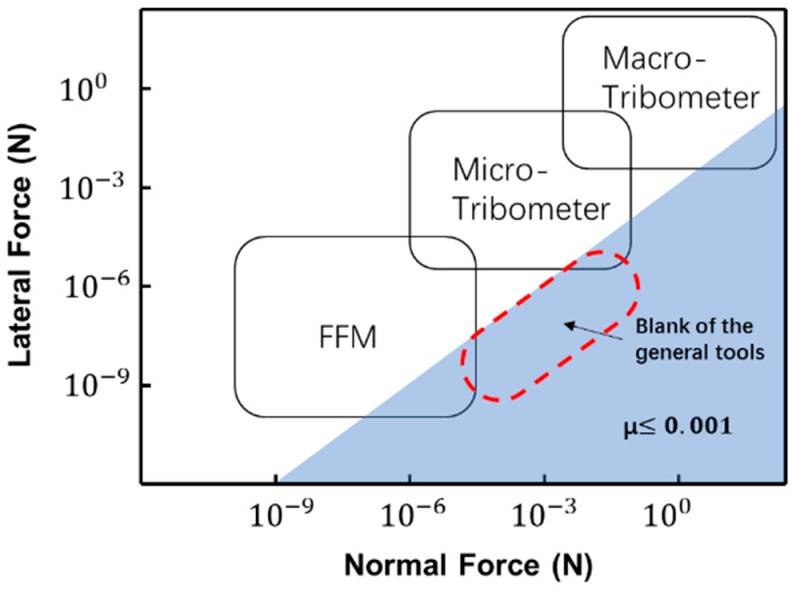
Summary of the current commercial instruments for tribology studies. The shade shows the requirements of forces in SSL studies. Force ranges of the instruments are enclosed in the rounded squares. There is a blank of the general tools which is circled by the dotted line.

**Figure 2 sensors-19-03431-f002:**
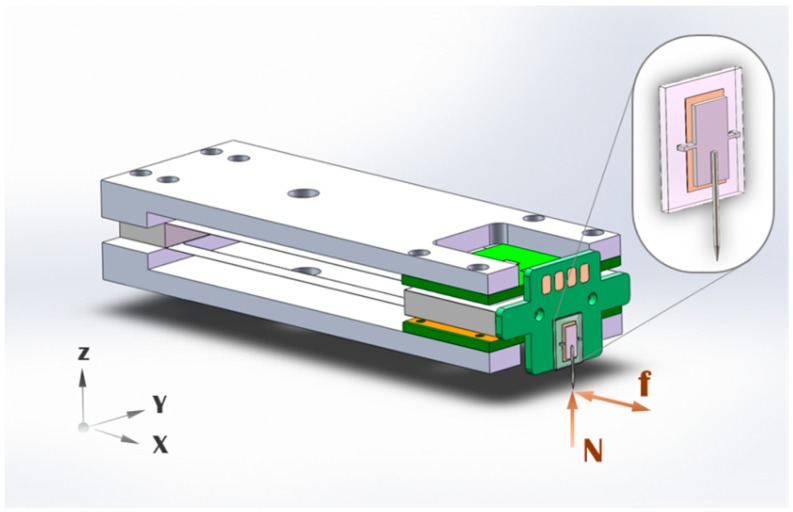
Schematic model of the hybrid two-axis force sensor.

**Figure 3 sensors-19-03431-f003:**
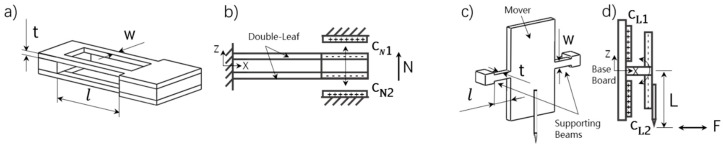
Schematic of sensing structures. (**a**,**b**) Structure for the normal load detection. (**a**) A 3D model of the double-(leaf-)cantilever pair; (**b**) a side view sketch of the differential capacitor configuration. (**c**,**d**) Structure for the lateral load detection. (**c**) A 3D model of the teeter-totter structure; (**d**) a side view sketch of the differential capacitor configuration.

**Figure 4 sensors-19-03431-f004:**
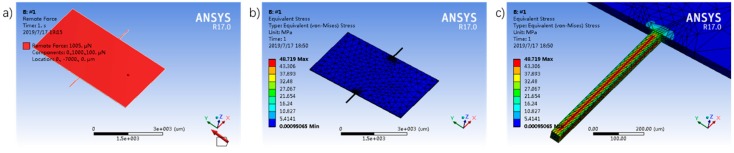
FEA on teeter-totter structure. (**a**) Loads applied on the structure—100 μN lateral force and 1 mN normal load; (**b**) meshing result, locally refined; (**c**) simulation result of the max equivalent stress.

**Figure 5 sensors-19-03431-f005:**
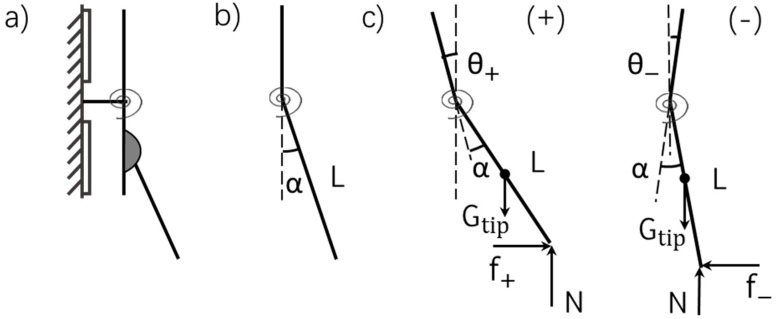
Geometric illustration of the coupling from the teeter-totter structure. (**a**) A side view model of the teeter-totter structure with a tip connected on the plate by glue; (**b**) A simplified model from (**a**); (**c**) Two common situations (forward and backward) during the friction measurements.

**Figure 6 sensors-19-03431-f006:**
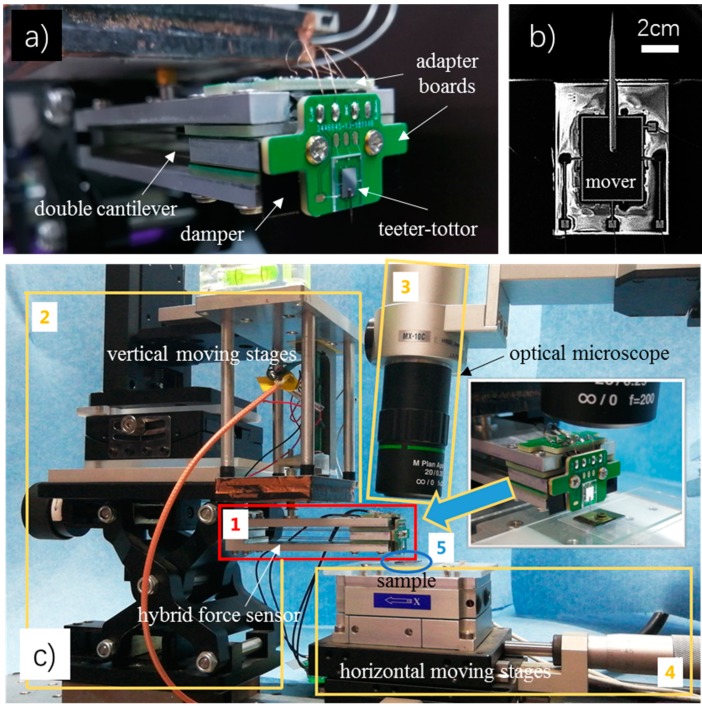
Photos of the sensor and the test system. (**a**) A photo of the hybrid two-axis force sensor; (**b**) A SEM picture of the micro teeter-totter sensing part; (**c**) A photo of the test system. 1-the hybrid force sensor, 2-vertical moving stages, 3-optical microscope (Hirox), 4-horizontal moving stages, 5-a test sample. The inset picture at the upper right corner is a close look at the microscope, the sample, the lateral force sensor and the adapter boards.

**Figure 7 sensors-19-03431-f007:**
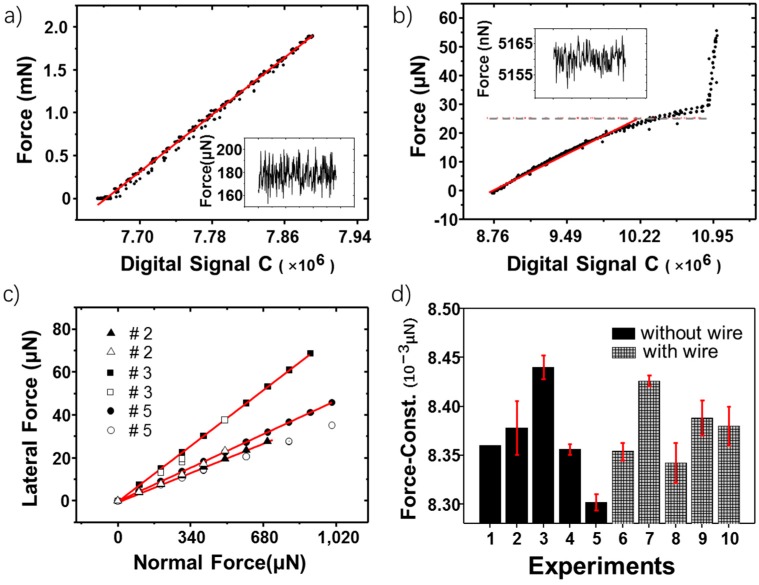
Calibration results. (**a**,**b**) Relationships between the digital capacitance signals and the reference forces. The inset picture in each picture shows the peak-to-peak noise. The red line is the curve fitting result. (**a**) The normal load sensing part; (**b**) the lateral force sensing part. The grey dotted line represents a force value of 25 μN, which is the recommended range for a better linearity. (**c**) Geometric coupling calibration. Filled icons are the theoretical results; hollow icons are the experimental results. The red line is the curve fitting. The diagram illustrates the relationship between the applied normal load and the induced lateral signal. (**d**) Assembly error tests. The black columns show the experiments without wires. The grey-grade columns show the experiments with wires. Each height of the column represents an average calibrated force constant ($△N/△C_{N}$) of each assembly. Error bars are shown to represent the std within each assembly.

**Figure 8 sensors-19-03431-f008:**
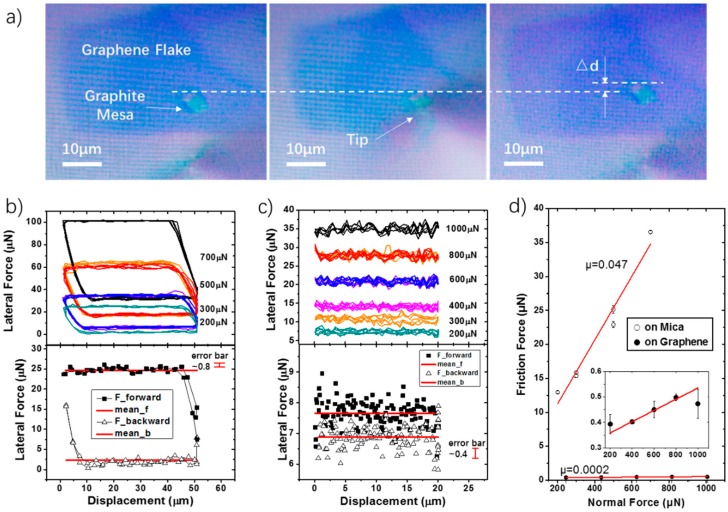
Friction experiments. (**a**) Photos observed by the optical microscope. The shoe-like shape in a dark purple color is the graphene flake; the blue square pointed out by an arrow is the 5 μm × 5 μm graphite mesa, the tip is shown in a bright triangle shape at the bottom of each photo. A small sliding distance of the graphite mesa can be observed after applying a load on it by the tip; (**b**) friction loops of the graphite-mica interface using the #3 teeter-totter beam; (**c**) friction loops of the graphite-graphene interface using the #5 teeter-totter beam. (**b**,**c**) The picture at the bottom shows a magnified friction loop under the smallest normal load in each case. Points labeled in black squares and white triangles represent signals of different moving directions. The red thick line is the average lateral force in each direction; (**d**) relationships between the normal forces and the friction forces. The insert diagram is a magnification of the graphite-graphene case.

**Table 1 sensors-19-03431-t001:** Mechanical characteristics of the sensing structures.

Structure		Thickness *t* (mm)	Width *w* (mm)	Length *l* (mm)	Distance *L* (mm)	Stiffness ^1^ *K* (N/m)
Double-Cantilever		0.2	35	56		127.55
Teeter-Totter	#1	0.04	0.04	0.7	7	1.30
	#2	0.04	0.04	0.4	5.6	3.57
	#3	0.04	0.04	0.4	6.2	2.91
	#4	0.04	0.08	0.7	7	4.24
	#5	0.04	0.12	0.7	4.9	14.90

^1^ Linear format stiffness.

**Table 2 sensors-19-03431-t002:** Calibration results of the lateral force sensing part and the available normal load of the corresponding hybrid sensors.

No.	$\mathrm{Stiffness}\;K_{F}\;(N/m)$	Range			Resolu. (nN)	Max Load (mN)
		θ (°)	*d* (μm)	*F* (μN)		
#2	3.38	±0.09	±9.38	±30	10	0.683
#3	4.98	±0.09	±10.04	±70	30	0.926
#5	13.61	±0.08	±7.34	±170	80	5.05

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
