# Peer review of "A Hybrid Two-Axis Force Sensor for the Mesoscopic Structural Superlubricity Studies"

_sensors, 2019, doi:10.3390/s19153431_

Round 1
Reviewer 1 Report
This manuscript describes the construction of a two-axis sensor that combines high lateral sensitivity with the ability to exert high pressure on mesoscopic scale surfaces.
The design of this sensor is well described and its manufacture, combining mechanical engineering and micro-manufacturing techniques, requires a certain know-how. Even if its implementation requires taking into account many couplings (mechanical, electronic...) the results obtained on the friction of graphene flakes are significant enough to validate the method. I recommend the publication of this manuscript but would appreciate it if the authors could consider the following remarks:
- If it had been shown that this sensor was able to make an AFM image of a sample it would have a very big advantage for publication.
- I would also like to know how the authors compare their system with the quartz tuning fork introduced by Prof. Franz Giessibl (see: https://doi.org/10.1063/1.5052264). It is also possible with this instrument to couple high pressures with high lateral sensitivity.
- In the same vein and closer to the philosophy of this manuscript. What do the authors think of the two-axis measuring system introduced very recently by Prof. Lydéric Bocquet as part of the friction measurement on ice (see: arXiv:1907.01316)?
Reviewer 2 Report
The reviewed manuscript reports on the development and evaluation of a hybrid two axis force sensor, designed for measuring friction and normal forces that differ by orders of magnitude. The sensor utilizes a double-leaf flexure assembly for normal force, and a torsional leaf flexure for lateral force detection; force transduction in both axes is capacitive. The authors provide theoretical background on the sensor design and fabrication and present a number of figures and sketches in support of such description. Although the elements of this design are not unique, the hybrid sensor itself is novel and provides a good basis for future sensor design and low-force tribometry. The manuscript is technically sound although it could be improved both technically and grammatically. I recommend some revisions prior to publication, per comments below:
1. Page 2, line 50: this sentence is misleading. The maximum normal load achievable in an AFM is of the order of 1 mN or more since this is highly probe-specific. For example, stainless steel cantilevers can easily reach high normal loads in an AFM, so can the probes reported in Garabedian et al. (which the authors cite). Furthermore, contacts in AFM routinely experience contact stresses in the vicinity of 10+ GPa. The choice of a 1um x 1um area is furthermore arbitrary. Most FFM contacts are significantly smaller (which, in part result in high stresses).
2. Page 2, line 70 and 76: The authors indicate that Garabedian et al.’s design “blocks the light to observe the contact mates”, and suggest later that “real-time observation of the contact mates has been achieved” (in the current work). While their claim of Garabedian’s work is accurate, their claim for the current work is inaccurate – the proposed sensor design does not intrinsically allow visualization of the contact pair and is not much different from prior AFM/FFM work in its inability of allow visualization. I recommend the authors temper such claims.
3. The plot in figure 1 is misleading. The colored bands, for example do not represent iso-lines. Additionally, the authors’ definition of the transition between nano and microscale is contrary to what is universally accepted for “nanoscale” (<100nm, or 0.01 um^2). Similarly, the claim on page 3 line 85 is arbitrary – the contact pressure/area conditions where superlubricity collapsed in ref. 6 is not necessarily a universal limit for all material systems. I recommend omitting this graph.
4. Page 5: for the text alluding to Figure 4, the authors should clearly annotate on the figure which segments of the structure are being referred to as “supporting beams”, “mover plate”, etc. The authors’ name designation for different segments of the sensor are not clear, which make it difficult to follow the analysis on page 5 and onward.
5. Per the previous point, it is unclear what the authors mean by “collapse deformation along the beams” (line 177) and “bending deformation of the lower part of the mover plate”. In accounting for all deformations, have the authors considered the bending deflection of the probe itself between the point of contact and its junction with the mover plate?
6. Table 1, “Thichkness” should read ‘thickness’
7. Table 1: What does “Distance, L” represent? This should be clearly annotated on the figure.
8. It appears that the authors account for effect on normal force on measured friction force (per section 3.2). Have the authors considered and included the torsional moment (due to friction), and indeed the bending deflection in the supporting beams of the teeter-totter influencing the measured normal force?
9. Figure 6(b): the authors should consider annotating this, and indeed other images in Figure 6 with clear descriptions of the various components.
10. The authors ascribe non-linearity in figure 7(b) to “inverse relationship between the capacitance value of the variable capacitor and the gap” (line 351). This is somewhat misleading since a similar inverse relationship exists for normal deflection, and indeed any capacitive sensing system, however there are no noticeable non-linearities in the normal deflection system. Furthermore, the authors contend that “to increase linearity, a better range to use is 25uN”. Again, this is misleading. Reducing the range does not make the response any more linear. I suggest the authors provide a more substantive discussion of the possible sources of this non-linearity (did they perform measurements with different flexures and notice trends, for example?). Is this response just non-linear but repeatable or is it non-linear and non-repeatable?
11. Page 10, the paragraph beginning at line 368 refers to “figure 7c”, which I believe should read “figure 7d”.
12. In figure 8, the authors report a friction coefficient of 0.0002 on graphene, with accompanying friction loops at various normal loads. This value seems aphysical especially considering how the friction loops appear at various loads. My guess is that the #5 flexure is too stiff to allow reasonable measurement of friction – it deflects too little to measure the ultra-low friction, resulting in nearly flat friction loops shown in Fig. 8c. Did the authors attempt these measurements on graphene with a softer flexure (say, #2)? In other words, it appears that the measurement against graphene is not sensitive to low interfacial friction. This is something that should improve with a softer lateral force flexure.
Reviewer 3 Report
In this paper, a hybrid sensor for simultaneous measurement of large normal force and small shear force was designed and tested, and the performance of the sensor was verified by experiments. The topic is well relevant to the journal, and some of the results are very interesting. I recommend that this paper can be accepted after making minor revisions. Here are some comments and suggested changes that need to be addressed:
1. In the sensor design part of this paper, the analysis of three deformation forms of the swing structure need to be briefly stated, which makes the whole calculation more convincing by using torsional stiffness instead of system stiffness.
2. The finite element analysis was used to simulate the stress state of micro teeter-totter structure. If possible, the analysis process or results need to be shown in the form of pictures. It is not enough to describe the analysis results only in the text.
3. Why only #2, #3 and #5 teeter-totter structures were involved in the analysis of signal coupling? And as a comparison, #5 teeter-totter structure has three variables relative to #2, #3.
4. In the introduction section, more recent relative work should be concerned, such as: Sensors 2019, 19(14), 3116; Wear 422–423 (2019) 201–211.
5. In the study of measurement uncertainty caused by assembly, the description of Fig. 7c is shown in this paper. It should be Fig. 7d that reflects the problem described.
6. Figure 7c shows a comparison between the theoretical and experimental results on the coupling effect. As a comparison, why are the points taken from the experimental and theoretical analysis inconsistent? The test value is less, and the normal force of three groups of teeter-totter structures is different.
7. In the friction test section, about 8b and c, why choose different teeter-totter beams in different friction interfaces? If it is a comparison, the same teeter-totter beam would be better.
8. Unfortunately, I do not find the supplementary materials in the system. The website www.mdpi.com/xxx/s1 cannot be open. Where are the Figures S1, S2, S4 and S5 mentioned in the article? And no Figure S3?
